# A Model Construction of Starvation Induces Hepatic Steatosis and Transcriptome Analysis in Zebrafish Larvae

**DOI:** 10.3390/biology10020092

**Published:** 2021-01-27

**Authors:** Hao Xu, Yu Jiang, Xiao-Min Miao, Yi-Xi Tao, Lang Xie, Yun Li

**Affiliations:** 1Institute of Three Gorges Ecological Fisheries of Chongqing, College of Fisheries, Southwest University, Chongqing 400715, China; xuhao@email.swu.edu.cn (H.X.); jjhehe@email.swu.edu.cn (Y.J.); mxm1999@email.swu.edu.cn (X.-M.M.); tyx0914@email.swu.edu.cn (Y.-X.T.); memcpy@email.swu.edu.cn (L.X.); 2Key Laboratory of Freshwater Fish Reproduction and Development (Ministry of Education), Key Laboratory of Aquatic Science of Chongqing, Southwest University, Chongqing 400715, China

**Keywords:** zebrafish, starvation, hepatic steatosis, NAFLD, *cd36*, RNA-seq

## Abstract

**Simple Summary:**

Nonalcoholic fatty liver disease (NAFLD) is the most common chronic liver disorder. Previous studies have focused on NAFLD caused by factors such as obesity or high-fat diets, but in recent years, more and more studies have proved that starvation is also an important cause of NAFLD. It has shown great harm in human health (e.g., dieting and anorexia nervosa) and farming economic benefits (e.g., laying hens and mink). However, the molecular mechanism underlying of starvation-induced NAFLD remain unclear. Regarding the study of NAFLD, zebrafish is currently widely used model organism. Thus, in this study, we used zebrafish to establish a starvation-induced hepatic steatosis model. As we all know, hepatic steatosis is usually a prerequisite for NAFLD. Subsequently, we performed a comparative transcriptome analysis of starvation-induced hepatic steatosis by RNA-Seq. We demonstrated that starvation triggers hepatic steatosis by promoting extrahepatic fatty acid uptake and lipogenesis, and inhibits hepatic fatty acid metabolism and lipid transport. Based on the indications provided by these data, we further revealed that *cd36* plays a crucial role in regulating extrahepatic fatty acid uptake and inducing hepatic steatosis during starvation conditions. Altogether, these findings will help us understand the pathogenesis of starvation-induced NAFLD.

**Abstract:**

Hepatic steatosis caused by starvation, resulting in non-alcoholic fatty liver disease (NAFLD), has been a research topic of human clinical and animal experiments. To understand the molecular mechanisms underlying the triggering of abnormal liver metabolism by starvation, thus inducing hepatic lipid accumulation, we used zebrafish larvae to establish a starvation-induced hepatic steatosis model and conducted comparative transcriptome analysis by RNA-seq. We demonstrated that the incidence of larvae steatosis is positively correlated with starvation time. Under starvation conditions, the fatty acid transporter (*slc27a2a* and *slc27a6-like*) and fatty acid translocase (*cd36*) were up-regulated significantly to promote extrahepatic fatty acid uptake. Meanwhile, starvation inhibits the hepatic fatty acid metabolism pathway but activates the de novo lipogenesis pathway to a certain extent. More importantly, we detected that the expression of numerous apolipoprotein genes was downregulated and the secretion of very low density lipoprotein (VLDL) was inhibited significantly. These data suggest that starvation induces hepatic steatosis by promoting extrahepatic fatty acid uptake and lipogenesis, and inhibits hepatic fatty acid metabolism and lipid transport. Furthermore, we found that starvation-induced hepatic steatosis in zebrafish larvae can be rescued by targeting the knockout *cd36* gene. In summary, these findings will help us understand the pathogenesis of starvation-induced NAFLD and provide important theoretical evidence that *cd36* could serve as a potential target for the treatment of NAFLD.

## 1. Introduction

Nonalcoholic fatty liver disease (NAFLD) is the most prevalent chronic liver disorder caused by lipid metabolism disorders, and is characterized by excessive lipid accumulation in hepatocytes [1]. A recent report showed that, in Asia, the estimated prevalence of NAFLD has reached 29.62% [2]. NAFLD is a relatively broad classification. According to the degree of pathological changes and whether the diseased liver tissue is accompanied by inflammation or fibrosis, NAFLD can be divided into: simple fatty liver, non-alcoholic steatohepatitis (NASH), and NASH-related cirrhosis [3]. Additionally, NAFLD can also be divided into primary and secondary types. Generally, primary NAFLD is often associated with metabolic syndromes such as obesity, insulin resistance (IR), type 2 diabetes mellitus (T2DM) and cardiovascular disease [4]; while secondary NAFLD is mainly caused by malnutrition, dieting, weight loss after bariatric surgery, and drug or toxic substance poisoning [5]. However, compared with the former, reports on secondary NAFLD are relatively scarce. Currently, although there is no detailed data on the relative prevalence of primary and secondary NAFLD, NAFLD caused by malnutrition and dieting is not uncommon.

It is well-known that the liver plays a central role in the maintenance of systemic lipid homeostasis during fasting and re-feeding cycles. Under the postprandial state, the liver uptakes dietary fats to synthesize triglycerides for very low-density lipoprotein (VLDL)-mediated secretion and transport to adipose tissue for storage, or transport to the heart and skeletal muscle for β-oxidation. During fasting, fatty acids are mobilized from adipocytes to the liver for energy supply. In the liver, the influxed fatty acids from extrahepatic organs are oxidized by β-oxidation pathway, leading to the production of acetyl coenzyme A (acety-COA), which then condenses with itself to ketone bodies (i.e., acetoacetic acid, β-hydroxybutyric acid, and acetone), thereby providing energy for other tissues, such as the brain, during starvation; and thus, the uptake of extrahepatic fatty acids has been considered a survival strategy in response to energy deficiency [6]. However, it has been documented that prolonged starvation can cause liver tissues to transition from simple hepatic steatosis to severe liver disease, including inflammation, cell death and fibrosis. For example, sudden anorexia in cats causes severe negative energy balance, accompanied by impaired liver function, intrahepatic cholestasis, and intrahepatic lipid deposition [7]. During weaning or fall moulting, minks lose their appetite for many days, resulting in rapid weight loss and lipid accumulation in the liver [8,9,10]. Furthermore, some case reports of dieting and anorexia patient also present serious NAFLD [11,12,13], which has become a potential risk problem for the public health of adolescents.

A metabolomic approach has been applied to a mouse model of starvation-induced hepatic steatosis [6]. After 24 h of fasting, compared with the control group, the free cholesterol (FC), cholesterol esters (CE) and triacylglycerols (TG) in the liver increased by 192%, 268%, and 456%, respectively. This indicates that, in animals, the lipid profile in the liver is a dynamic system, which is readily affected by environmental factors such as starvation. Nevertheless, the underlying mechanisms of hepatic steatosis caused by starvation is largely unknown. In addition to the influx of a large number of extrahepatic fatty acids, β-oxidation, de novo lipogenesis and lipid transport are the vital factors affecting hepatic steatosis in lipid metabolism during starvation. In the liver, increased fatty acid uptake and de novo lipogenesis, in addition to decreased β-oxidation and VLDL secretion, can all cause hepatic steatosis. However, it is still unclear which lipid metabolism abnormalities are caused by starvation and result in liver steatosis.

In recent years, the use of zebrafish as a model organism for the study of NAFLD has been well established, including diet-induced, chemical-induced, and transgenic models [14,15,16,17]. The advantages of the use of zebrafish are rapid development, a short spawning cycle, large number of eggs, optical transparency, easy observation, and low financial cost. Thus, in this study, we used zebrafish to establish a starvation-induced hepatic steatosis model and performed comparative transcriptome analysis by RNA-seq. We also used CRISPR/Cas9 genome editing system to establish a homozygous fatty acid translocase (*cd36*) knockout zebrafish. Our data revealed that starvation induces hepatic steatosis by promoting extrahepatic fatty acid uptake and lipogenesis, and inhibiting hepatic fatty acid metabolism and lipid transport. Via suppressing the role of the *cd36* gene, the steatosis in zebrafish larvae can be rescued. Our aim was to provide an insight into the molecular mechanisms underlying starvation-induced hepatic steatosis, which will help improve the understanding of NAFLD caused by malnutrition. Furthermore, we believe that our findings have important implications for broadening the study of secondary NAFLD.

## 2. Materials and Methods

### 2.1. Animals and Starvation Treatment

Zebrafish (*Danio rerio*) were reared under standard laboratory conditions with a light/dark cycle of 14/10 h at 28.5 °C. Embryos were collected and maintained in embryo medium. At 5 dpf, larvae were randomly divided into two groups. Each group included three biological replicates, and 40 embryos as one biological sample. One group, as the controls, were fed with 15 mg AP100 diets (Haisheng, China) once per day. The other, as the starvation group, was not provided with the basal diets, i.e., fasting treatment. In this study, livers were collected daily from both the controls and starvation group for relevant studies until 10 dpf. The animal experiments were conducted according to the regulations of the Guide for Care and Use of Laboratory Animals, and they were approved by the Committee of Laboratory Animal Experimentation at Southwest University (IACUC Issue No. 2019071806; Approval date: 18 July 2019).

### 2.2. Oil Red O (ORO) Staining

Hepatic neutral lipids were labeled by ORO staining, as described previously [15]. Briefly, zebrafish larvae were fixed in 4% paraformaldehyde (PFA) overnight at 4 °C, and were then washed twice with phosphate-buffered saline with 0.1% Tween-20 (PBST). Next, samples were incubated into a filtered solution of 0.3% ORO in 60% isopropanol for one hour. The samples were then rinsed twice with 60% isopropanol for 5 min each time and repeatedly washed with PBST. Stained larvae were stored in 4% PFA and the representative images were obtained using a LEICA DM 3000 microscope (Leica, Wetzlar, Germany).

### 2.3. Electron Microscopy

After MS-222 anesthesia, we dissected the liver tissues and fixed them immediately in 2.5% glutaraldehyde at 4 °C. Then, the fixed livers were mounted, sectioned, and imaged, as previously described [18].

### 2.4. Biochemical Analysis

Triglycerides (TG) and very low-density lipoprotein (VLDL) in zebrafish livers were quantified using a Triglyceride Assay Kit (Nanjing Jiancheng Bioengineering Institute, Nanjing, China) and Fish VLDL Elisa Kit (BIOHJ, Xiamen, China) according to the manufacture’s specifications.

### 2.5. Quantitative RT-PCR

Total RNA was isolated from the dissected liver tissues of zebrafish larvae with RNAiso plus reagent (Takara, Tokyo, Japan), and RNase-free DNase was employed to remove the contaminating DNA; cDNA was synthesized from 2 μg of total RNA with GoScript^TM^ Reverse Transcription System (Promega, Madison, WI, USA). qPCR assay using three replicates were performed using the CFX96 Real-Time PCR detection system (Bio-Rad, Hercules, CA, USA). All primers used are listed in Appendix A. All procedures were performed as previously described [19].

### 2.6. RNA-Seq and Transcriptomic Analysis

Zebrafish larvae were anesthetized by MS-222, and then the liver was dissected using a micro-dissection needle. Forty liver tissues were pooled into a tube as one sample. Total RNA extraction of three independent biological replicates were performed as stated above. RNA quality and concentration were assessed using Agilent 2100 Bioanalyzer (Agilent Technologies, Santa Clara, CA, USA). For high-throughput sequencing, the library preparations were conducted using Truseq^TM^ RNA Sample Preparation Kit (Illumina, San Diego, CA, USA) and sequenced on an Illumina Hiseq X10 platform by Majorbio Biopharm Technology Co., Ltd. (Shanghai, China).

After sequencing, the high-quality clean reads were obtained by filtering the dirty reads with the low-quality reads, adaptors, and sequences with a high content of N or reads <20 bp using SeqPrep and Sickle, and then mapped to the reference genome zebrafish (GRCz11) using Tophat [20]. The gene expression levels were calculated using the method of FPKM (Fragments Per Kilobases Per Millionreads) [21]. DESeq2 was used for the differential expression analysis of RNA-Seq expression profiles [22], and genes with a more than two-fold changes, and *p* < 0.05 and false discovery rate (FDR) < 0.05, were considered to be differentially expressed. The raw data has been uploaded to the NCBI Sequence Read Archive under the accession number PRJNA682774.

### 2.7. Cd36 Knockout by CRISPR/Cas9

To obtain the zebrafish *cd36^−/−^* mutant line, a specific CRISPR/Cas9 target site was designed using the target site prediction website (http://zifit.partners.org/ZiFiT/). After injecting cas9 mRNA and gRNA into ond-cell embryos, the knockout effect at 72 h post fertilization (hpf) was determined by pairs of specific primers (*cd36*exon1-F/R) using Sanger sequencing, as shown in Appendix A. The F0 generation positive mutants were raised to sexual maturity and then mated with WT zebrafish to obtain F1 generation mutants. Then, F1 generation mutants carrying the same genotype were used for mating to obtain F2 homozygous mutants. In this study, the polyacrylamide gel electrophoresis (PAGE) was employed to rapidly identify the different genotypes of F2 generation mutants [23].

### 2.8. Statistical Analysis

Data are expressed as means ± SEM. One-way ANOVA with Tukey’s post-hoc test was performed for multiple group comparisons. For all analysis, *p* < 0.05 was considered statistically significant.

## 3. Results

### 3.1. Starvation Induced Hepatic Steatosis in Zebrafish Larvae

In this work, to confirm whether zebrafish larvae can be used to build a starvation-induced hepatic steatosis model, we performed a fasting treatment at 5 dpf, which is the first feeding stage of zebrafish larvae. At this stage, without timely supplementation of exogenous nutrients, the larvae will immediately enter a malnourished state. As shown in Figure 1A, the growth of the starved larvae was arrested completely during the starvation process. Commencing on 10 dpf, the starved larvae died intensively (Figure 1B), therefore we focused on the observation of hepatic steatosis in larvae from 5 to 10 dpf. ORO staining showed starvation resulted in a significant accumulation of neutral lipids in larval liver tissues. Moreover, the incidence of hepatic steatosis was positively correlated with the starvation time and reached its highest level three days after starvation (Figure 1C,D). By measuring the level of triglycerides (TG), we found a significant increase in TG levels in the livers of starved larvae compared to those of the control larvae (Figure 1E). Furthermore, by transmission electron microscope observation, we examined lipid droplets in the cytoplasm. In contrast, no lipid droplets were observed in the hepatocytes of larvae fed normally (Figure 1F).

### 3.2. Comparative Transcriptome Analysis of Starvation Induces Hepatic Steatosis

To clarify the molecular mechanism underlying hepatic steatosis caused by starvation in zebrafish larvae, we employed RNA-seq technology to investigate the effects of starvation on the gene expression regulation in the liver tissues. In our study, three biological replicates were performed in the control and starved larvae, respectively. The library construction, sequencing data, and quality control are shown in Table 1. In total, these results indicate that the RNA-seq data are of high quality and can be used for further analysis.

Correlation analysis between samples shows that the intra-group correlation coefficients in the control larvae and starved larvae exceeded 0.977 and 0.972, respectively (Figure 2A). This indicates that the variation in gene expression between intra-group samples was extremely low. In contrast, the correlation coefficient between inter-group samples was relatively low (0.822–0.908). This indicates that the gene expression of liver tissues changed significantly during starvation stress.

In this study, genes with differential expression levels ≥2 and *p*-adjust < 0.05 were considered to be significantly different. On the basis of this condition, 4166 differentially expressed genes (DEGs), including 2049 up-regulated and 2117 down-regulated, were identified (Figure 2B). Interestingly, among these DEGs, 4129 expressed in both the control and starved larvae. In addition, 28 genes were expressed only in the control and nine in the starved larvae alone (Figure 2C and Appendix A). Then, all of the DEGs with significant change were subject to further Kyoto Encyclopedia of Genes and Genomes (KEGG) enrichment analysis.

As shown in Figure 2D and Appendix A, starvation stress resulted in more than 20 pathways being considerably affected. As expected, pathways (i.e., DNA replication and cell cycle) involved in growth regulation were inhibited significantly. In addition, most of these pathways are associated with lipid metabolism, including fatty acid degradation, glutathione metabolism, peroxisome proliferator-activated receptor (PPAR) signaling pathway, fatty acid elongation, fat digestion, and absorption, biosynthesis of unsaturated fatty acids, and alpha-Linolenic acid metabolism. Thus, it appears that starvation stress has a significant effect on the hepatic lipid metabolism of zebrafish larvae.

### 3.3. Expression of Genes Related to Fatty Acid Transmembrane Transport in Liver Tissues of Starved Larvae

Under starvation conditions, liver tissues mainly uptake free fatty acids from the extrahepatic organs, and then synthesize triglycerides, phospholipids, and cholesterol esters in the hepatocytes, thereby providing major energy resources for tissues (e.g., heart, skeletal muscle, and brain). However, the influx of fatty acids into hepatocytes, with the exception of a small part (less than 10 carbon chains), is via passive diffusion, and more than 90% of long-chain fatty acids (LCFAs) require the transport of specific membrane proteins to enter [24]. It has been confirmed that several proteins, including fatty acid translocase (FAT/CD36) and fatty acid transporters (FATPs), play an important role in the transmembrane transport of fatty acids [25,26,27,28]. In this study, we identified six fatty acid transporters (*slc27a1a*, *slc27a1b*, *slc27a2a*, *slc27a4*, *slc27a6*, and *slc27a6-like*) (Figure 3A) and one fatty acid translocase (*cd36*) in the liver tissues of zebrafish. Among these transporters, *slc27a1b*, *slc27a4*, and *slc27a6* were down-regulated significantly, whereas *slc27a2a* and *slc27a6-like* were up-regulated. Notably, the expression level of *slc27a2a* was the highest among these six paralogs. In addition, we observed that *cd36* was also up-regulated (Figure 3B). These data suggest that the liver tissues of zebrafish larvae may mediate the transmembrane transport of *slc27a2a*, *slc27a6-like*, and *cd36*, promoting the uptake of extrahepatic fatty acids during starvation stress.

### 3.4. Expression of Genes Involved in Intracellular Fatty Acid Transport and β-Oxidation in the Liver Tissues of Starved Larvae

In liver, the influxed extrahepatic fatty acids are either used for fatty acid metabolism or de novo lipogenesis. Fatty acid binding proteins (FABPs) is a family of low molecular weight intracellular proteins involved in the transport and metabolism of fatty acids [29]. In the RNA-seq data, we detected 11 FABPs genes expressed in the liver tissues of zebrafish larvae (Figure 4A). Among these FABPs genes, we observed that four genes (*fabp1b.1*, *fabp1b.2*, *fabp2*, and *fabp7a*) were down-regulated significantly compared to control larvae (Figure 4B). Interestingly, however, the expression of *fabp10a* (fatty acid binding protein 10a, liver basic), which has the highest expression levels in the liver tissues, did not change significantly. Thus, it appears that *fabp10a* plays a vital role in regulating hepatic fatty acid intracellular transport during starvation stress.

The inhibition of a fatty acid metabolism pathway indicates that the influxed extrahepatic fatty acids do not appear to be used for β-oxidation in the liver tissues. As shown in Figure 5, a large number of the DEGs involved in the fatty acid metabolism were down-regulated significantly in the liver tissues of starved larvae. These genes include *cpt1ab* (carnitine palmitoyltransferase 1Ab, liver), *cpt1b*, *cpt2*, *acox1* (acyl-CoA oxidase 1, palmitoyl), *acox3*, *acsbg1* (acyl-CoA synthetase bubblegum family member 1), *acsl3a*, *acsl4a*, *acaa1* (acetyl-CoA acyltransferase 2), *acaa2*, *acadl* (acyl-CoA dehydrogenase long chain), *acadm*, *acadvl*, *acads*, *hadhaa* (hydroxyacyl-CoA dehydrogenase trifunctional multienzyme complex subunit alpha a), *hadhab*, *hadh*, *hadhb*, *eci1* (enoyl-CoA delta isomerase 1), *eci2*, *acat2* (acetyl-CoA acetyltransferase 2), *aldh3a2b* (aldehyde dehydrogenase 3 family, member A2b), and *aldh9a1b*. These data suggest that the influxed extrahepatic fatty acids is more tended to be de novo lipogenesis, rather than β-oxidation.

### 3.5. Expression of Genes Related to De Novo Lipogenesis in the Liver Tissues of Starved Larvae

Because starvation stress triggers the transmembrane transport of extrahepatic fatty acids but inhibits the intrahepatic fatty acid metabolism pathway in our above studies (Figure 3 and Figure 5), we therefore speculate that most of the fatty acids may be used for de novo lipogenesis rather than fatty acid metabolism. In the present study, we investigated the expression of major transcription factors and key enzymes involved in regulating de novo lipogenesis (Figure 6). These genes include peroxisome proliferator-activated receptors (PPARs), sterol regulatory element binding transcription factor (SREBFs), retinoid X receptor (RXRs), diacylglycerol O-acyltransferase (DGATs), stearoyl-CoA desaturase (SCD), and fatty acid synthase (FASN). Among these genes, we only detected two genes whose expression was down-regulated significantly. Conversely, the expression of most genes was not affected (e.g., *pparaa*, *pparg*, *rxraa*, *dgat2*, and *fasn*) by starvation, and three genes (i.e., *pparab*, *rxrgb*, and *scdb*) were even up-regulated. Thus, it appears that starvation stress promotes the lipogenesis of liver tissues in zebrafish larvae to a certain extent.

### 3.6. Lipid Transport in the Liver Tissues of Starved Larvae

Very low-density lipoprotein (VLDL) is essential for the transport of triglycerides from the liver into the circulation. It is well-known that apolipoproteins are a key component of VLDL, and have abundant subtypes expressed in liver [30]. In our study, 22 apolipoprotein genes with expressiosn ranging from 0.24 to 20773.21 FPKM were identified (Figure 7A). The top 15 genes were *apoa1b*, *apoa2*, *apoa4b.1*, *apoea*, *apoc1*, *apoc2*, *apo3b*, *apoa4b.2*, *apobb.1*, *apom*, *apoa4a*, *apoba*, *apodb*, *apoa1a*, and *apoc4* (Figure 7C). Notably, we observed that most genes were down-regulated significantly, with the exception of *apom* and *apodb*. This observation is consistent with the change of VLDL secretion in the liver of starved larvae. As shown in Figure 7B, we detected that the VLDL content in the liver of starved larvae was 47.54% lower than that of the control larvae. These results indicate that starvation may lead to lipid transport dysfunction in the liver of zebrafish larvae.

### 3.7. qPCR Analysis of Genes Related to Lipid Metabolism

To further verify the effects of starvation on the gene expression involved in hepatic lipid metabolism, we examined the relative mRNA levels for 20 genes by qPCR. It is found that the qPCR results of most candidate genes are highly consistent with the RNA-seq data (Figure 8A–D). These genes include up-regulated genes *cd36*, *slc27a2a*, and *scdb*, and down-regulated genes *fabp1b.1*, *cpt1ab*, *cpt1b*, *cpt2*, *acox1*, *acaa1*, *hadhab*, *apoa1*, *apoa4.3*, *apoba*, *dgat1a*, and *dgat1b*. In the qPCR data, only the expression of four genes (i.e., *dgat2*, *srebf1*, *srebf2*, and *fasn*) differed from the RNA-seq data. In brief, they were shown to be significantly up-regulated in the qPCR data, whereas no significant changes in the RNA-seq data were observed. Through a linear regression analysis, we determined that the fold change of the gene expression ratio between qPCR and RNA-seq data is highly positively correlated (R2 = 0.844) (Figure 8E). Thus, our RNA-seq data is reliable and of high quality.

### 3.8. Starvation-Induced Hepatic Steatosis Is Rescued by Targeting Knockout Cd36 Gene

Based on the above observations (Figure 8F), we hypothesize that the initial cause of starvation-induced hepatic steatosis may be triggered by excess fatty acid uptake in the liver. Our data from the RNA-seq and qPCR both showed that *cd36* (fatty acid translocase) was up-regulated the in the liver tissues of starved larvae (Figure 3B and Figure 8A), and is a major mediator of hepatic fatty acid uptake, as shown in previous reports [31,32,33]. Thus, to confirm this hypothesis, we employed CRISPR/Cas9 technology to target the knockout of the *cd36* gene, thereby blocking the pathway for the liver to uptake extrahepatic fatty acids under starvation conditions.

As shown in Figure 9A, the target site for gene knockout was designed in the exon1 of the *cd36* gene. Seventy-two hours after injection of gRNA and cas9 mRNA, DNA sequencing of normally developing F0 embryos revealed a high-frequency somatic mutation, and all mutations occurred correctly at the target sites (Appendix A). Upon PCR and sequencing, two homozygous mutants, 8-bp and 4-bp deletion in the exon1, were obtained from F2 generation (Figure 9B). Concurrently, individuals with different genotypes were screened by the polyacrylamide gel electrophoresis (PAGE) assay (Figure 9C). Ultimately, the mutants resulted in premature termination of translation and produced two truncated CD36 proteins (M1 and M2).

The mutants were fertile and no significant features of abnormality in the gross morphology between WT and *cd36^−/−^* mutants were observed (data not shown). Next, we performed ORO staining to investigate the effect of starvation on the hepatic steatosis of different genotypes in zebrafish larvae. As shown in Figure 9F,G, after three days of starvation, the hepatic lipid droplets of WT zebrafish larvae increased significantly and the percentages of hepatic steatosis reached 92.12%, whereas the steatosis degree of *cd36^−/−^* mutants did not appear to have significant changes before and after starvation. Consistently, after five days of starvation, only 22.83% of *cd36^−/−^* mutants showed hepatic steatosis, whereas the proportion of WT zebrafish reached 93.33% at the corresponding time. Through lipid content determination, we found that the hepatic lipid content of starved mutants was 44.09% lower than that of WT zebrafish (Figure 9H). These results demonstrate that starvation-induced hepatic steatosis is rescued by targeting the knockout *cd36* gene.

## 4. Discussion

In the present study, we successfully established a model of starvation-induced hepatic steatosis in zebrafish larvae. Furthermore, we examined the effect of starvation on gene expression regulation in the liver tissues by RNA-seq. Our data demonstrated that starvation contributes to hepatic steatosis by promoting extrahepatic fatty acid uptake and de novo lipogenesis, inhibiting hepatic fatty acid metabolism and lipid transport. Based on the indications provided by these data, we further revealed that *cd36* plays a crucial role in regulating extrahepatic fatty acid uptake during starvation conditions. To the best of our knowledge, this is the first animal model used to study the production of NAFLD by starvation in a non-mammalian vertebrate.

In our study, we defined a standard food intake, that is, 15 mg of diets per day for 40 fish larvae. Under this standard, the percentage of hepatic steatosis was less than 10%. In contrast, during food deprivation, we observed severe lipid accumulation in the liver of zebrafish larvae. Therefore, using this method of feeding is beneficial to identify the effects of starvation stress on the hepatic steatosis in zebrafish larvae.

At present, hepatic steatosis caused by starvation, thereby resulting in NAFLD, has been shown in human clinical [11,12,13,34,35] and animal experiments [6,7,8,36,37,38], but the molecular mechanisms underlying starvation-induced hepatic steatosis are yet to be well defined. Thus, we elucidated the regulation of gene expression in starvation-induced hepatic steatosis in this work. In the RNA-seq data, we observed significant changes in the expression of more than 4000 genes, including 2049 up-regulated genes and 2117 down-regulated genes. It is well-known that the excessive accumulation of lipids in hepatocytes under starvation stress is nothing more than (1) increased fatty acid flux to the liver from extrahepatic organs, (2) inhibited β-oxidation, (3) increased de novo lipogenesis, and (4) repressed lipid export [39]. Based on the above indications, we therefore focused on the expression of genes related to hepatic lipid metabolism. In the RNA-seq data, we first investigated the expression of genes involved in the transmembrane transport of fatty acids. Two fatty acid transport protein genes (*slc27a2a* and *slc27a2a-like*) and one fatty acid translocate gene (*cd36*) were shown to be up-regulated significantly. Previous reports have been shown that zebrafish *slc27a2a* is mainly expressed in the liver tissues of zebrafish [30,40]. Consistently, our data revealed that the expression of *slc27a2a* in the FATPs family is highest in the liver. CD36, also known as fatty acid translocase (FAT), is a fatty acid transporter that facilitates fatty acid uptake and has a profound impact on the development of NAFLD. Increased expression of hepatic CD36 leads to lipid accumulation, whereas inhibition of CD36 is resistant to hepatic steatosis [33,41]. Our data suggest that starvation may mediate fatty acid transmembrane transport of *slc27a2a*, *slc27a2a-like*, and *cd36*, thereby promoting the extrahepatic fatty acid uptake.

The KEGG enrichment analysis showed that starvation is involved in the regulation of a large number of pathways. Among these, the hepatic lipid metabolism related pathways, including the fatty acid metabolism, PPAR signaling pathway, fatty acid elongation, fat digestion and absorption, and biosynthesis of unsaturated fatty acids shows significant changes. Notably, we found that all DEGs enriched in the fatty acid metabolism pathway (e.g., *cpt1ab*, *cpt1b*, *cpt2*, *acox1*, *acaa1*, and *hadhab*) are down-regulated. This result indicates that although starvation promotes the transmembrane transport of fatty acids in the liver, the influxed fatty acids do not appear to be used for β oxidation. In contrast, we observed that several key genes (i.e., *srebf1* [42], *srebf2*, *fasn* [43], and *scdb* [44]) involved in de novo lipogenesis were up-regulated by qPCR, and the expression level of most other genes related to lipogenesis is not affected by starvation in the RNA-seq data. Thus, it appears that decreased hepatic fatty acid oxidation and increased lipogenesis are the vital causes of severe hepatic steatosis in the starved zebrafish larvae.

Concomitantly, we also observed that lipid transport is significantly impaired. Apolipoprotein is a protein component that constitutes plasma lipoprotein. Its basic function is to carry lipids and stabilize the structure of lipoproteins (e.g., VLDL) [45]. In the RNA-seq data, 22 apolipoprotein genes such as ApoA-I (*apoa1a* and *apoa1b*), ApoA- II (*apoa2*), ApoA-IV (*apoa4a*, *apoa4b.1*, *apoa4b.2*, and *apoa4b.3*), ApoB (*apoba*, *apobb.1*, and *apobb.2*), ApoC (*apoc1*, *apoc2*, and *apoc4*), and ApoE (*apoea* and *apoeb*) families were identified. However, most genes were shown to be down-regulated significantly in the liver of starved larvae. Furthermore, we also demonstrated that VLDL secretion from the liver is inhibited during starvation stress, which is consistent with a previous report by Gibbons et al. [46]. Thus, the other vital causes for starvation-induced hepatic steatosis in zebrafish larvae is the impairment of hepatic lipid transport function.

Previous reports have shown that the CD36 pathway plays a crucial role in NAFLD induced by high-fat diets [47,48,49,50], which is a key transporter of free fatty acid uptake in liver. In high-fat diet induced NAFLD patients and animal models, up-regulation of CD36 were commonly detected. On the contrary, suppressing its expression can effectively improve hepatic lipid accumulation. However, it is still unclear whether it plays a similar role during the starvation process, although our data show that starvation induces a significant up-regulation of *cd36* in the liver of zebrafish larvae. [14] has shown that starvation can regulate hepatic lipid metabolism through PPAR signaling pathway. Meanwhile, PPARγ, a positive regulator of CD36, is a transcriptional target of Pregnane X receptor (PXR) [51]. In our work, we examined that the expression of multiple key transcription factors involved in PPAR signaling pathway were not inhibited (e.g., *pparaa*, *pparg* and *rxraa*) by starvation, and two (i.e., *pparab* and *rxrgb*) were even up-regulated. Thus, it appears that starvation may mediate the PPAR signaling pathway to induce CD36 expression. To further explore the role of *cd36* in starvation-induced hepatic steatosis, we successfully created two independent mutant lines using the CRISPR/Cas9 system in zebrafish. Under starvation conditions, we observed the hepatic lipid droplets of WT zebrafish larvae increased significantly, whereas the steatosis degree of *cd36^−/−^* mutants did not appear to have significant changes before and after starvation. The comparison analysis showed that the hepatic lipid content of starved *cd36^−/−^* mutants is 44.09% lower than that of WT zebrafish. These results revealed the importance of *cd36* in regulating starvation-induced hepatic steatosis in zebrafish larvae.

## 5. Conclusions

In summary, our data suggests that starvation-induced hepatic steatosis is caused by complex lipid metabolism disorders. These events include increased hepatic fatty acids uptake and de novo lipogenesis, and inhibited β-oxidation and lipid export. We also demonstrated that the degree of hepatic steatosis in animals can be reduced by targeting the inhibition of the expression of *cd36* under starvation conditions. These findings will extend our understanding of abnormal liver lipid metabolism under starvation stress and clarify the intervention effect of *cd36* as a target, which may provide new strategies for the treatment of NAFLD.

## Figures and Tables

**Figure 1 biology-10-00092-f001:**
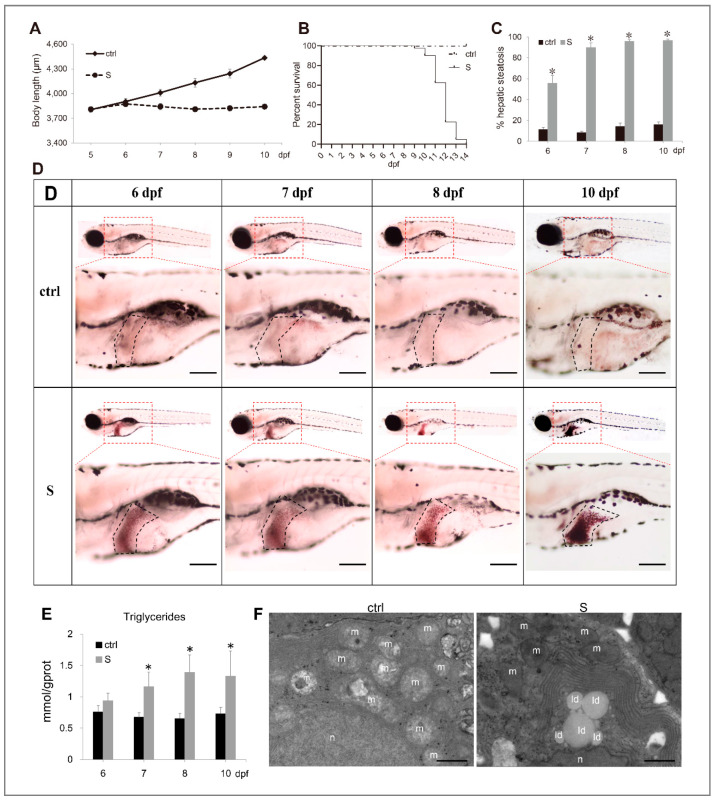
Characterization of starvation-induced hepatic steatosis in zebrafish larvae. (**A**) The depiction of growth performance in the control and starved larvae from 5 to 10 dpf. (**B**) The survival curve of the control and starved larvae was depicted by Kaplan-Meier assay. (**C**) Percentages of control and starved larvae with hepatic steatosis at 6, 7, 8, and 10 dpf. (**D**) Representative images of whole-mount Oil Red O (ORO) staining labelled the neutral lipids in larval liver at 6, 7, 8, and 10 dpf. The black dotted box labels the liver of zebrafish larvae. Bars = 200 μm. (**E**) Hepatic triacylglycerol (TAG) levels in the control and starved larvae from 6 to 10 dpf were determined. (**F**) Electron micrographs of hepatocytes from the control and starved larvae. Id: lipid droplets; m: mitochondria; n: cell nucleus. Bars = 10 μm. * indicates significant differences (*p* < 0.05).

**Figure 2 biology-10-00092-f002:**
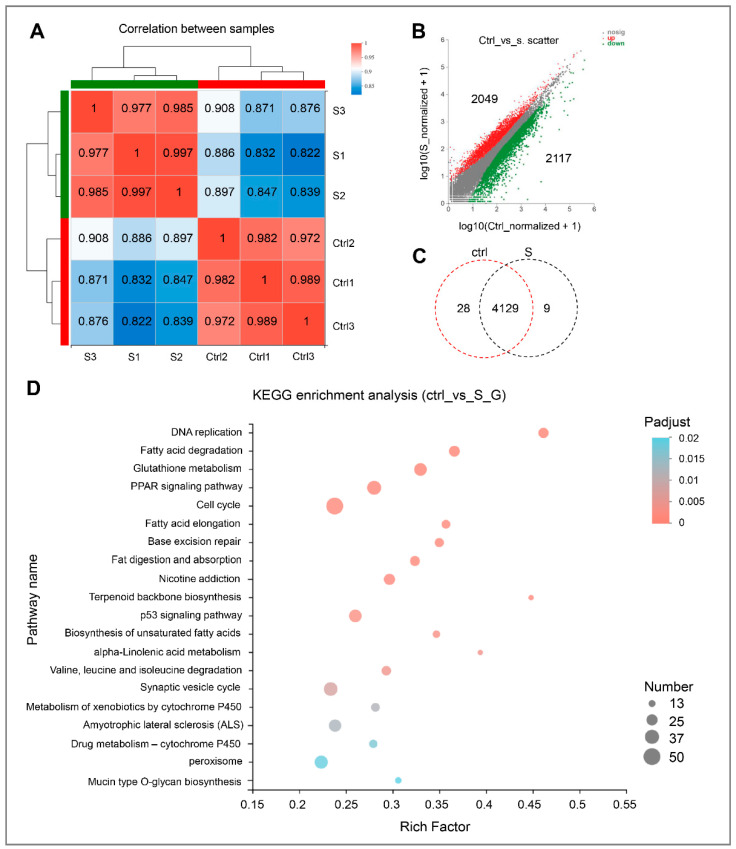
Gene expression and functional enrichment analysis of RNA-seq data. (**A**) Correlation analysis of intra-group and inter-group samples, respectively. (**B**) Scatter plot of differentially expressed genes in the control and starvation larvae. The red dots indicate up-regulated genes, and green dots indicate down-regulated genes. Ctrl represents the control group, that is, normal feeding larvae; S represents the starvation group. (**C**) Venn diagram analysis of differentially expressed genes in the control and starved larvae. (**D**) Scatter plot for KEGG enrichment analysis between control and starved larvae. The size and color of the circles represent the number of DEGs and the *p*-adjust value, respectively.

**Figure 3 biology-10-00092-f003:**
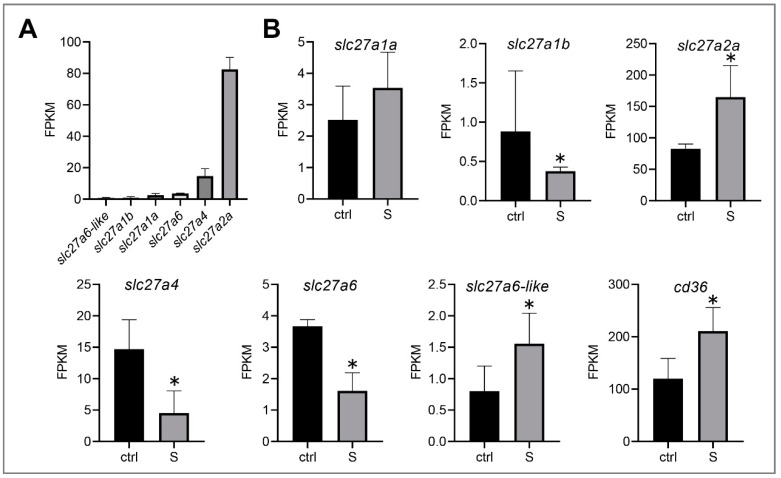
Effect of starvation on the gene expression involved in the transmembrane transport of fatty acids in liver tissues. (**A**) Gene expression of solute carrier family 27 (fatty acid transporter), including *slc27a1a*, *slc27a1b*, *slc27a2a*, *slc27a4*, *slc27a6*, and *slc27a6-like*, were extracted in the control larvae from the RNA-seq data. (**B**) Differential expression of fatty acid transporter and fatty acid translocase (*cd36*) in the control and starved larvae. The FPKM (Fragments Per Kilobases Per Millionreads) value represents the mean ± SEM of three independent RNA-seq biological replicates. * indicates significant differences (*p* < 0.05).

**Figure 4 biology-10-00092-f004:**
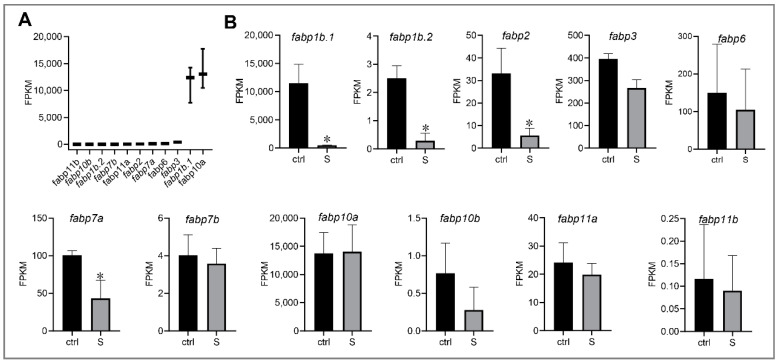
Effect of starvation on the gene expression involved in fatty acid binding proteins (FABPs) in liver tissues. (**A**) Gene expression of FABPs, including *fabp1b.1*, *fabp1b.2*, *fabp2*, *fabp3*, *fabp6*, *fabp7a*, *fabp7b*, *fabp10a*, *fabp10b*, *fabp11a*, and *fabp11b*, were extracted in the control larvae from RNA-seq data. (**B**) Differential expression of FABP genes in the control and starved larvae. The FPKM value represents the mean ± SEM of three independent RNA-seq biological replicates. * indicates significant differences (*p* < 0.05).

**Figure 5 biology-10-00092-f005:**
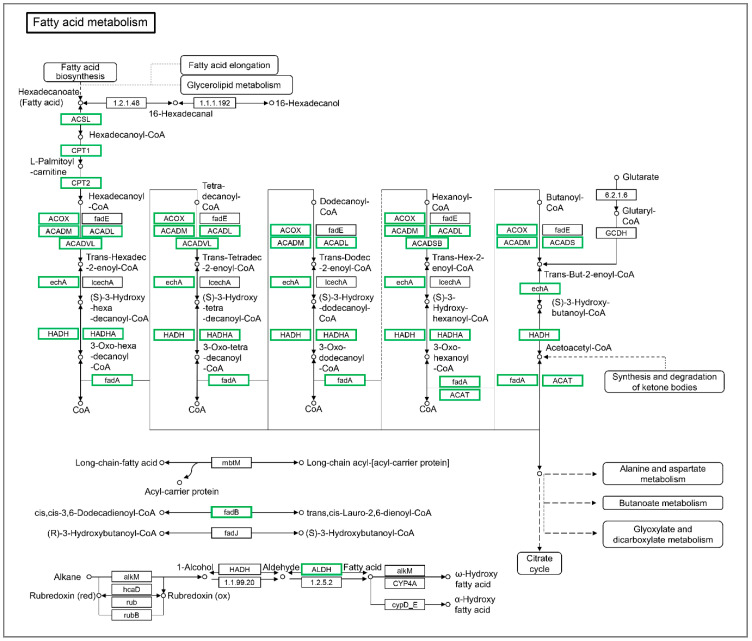
The differentially expressed genes identified by KEGG in the fatty acid metabolism pathway. The DEGs included *BX890602.1*, *acsl3a*, acsl4a, *acsbg1*, *cpt1ab*, *cpt1b*, *cpt2*, *acox1*, *acox3*, *acadm*, *acadl*, *acadvl*, *acads*, *hadhaa*, *hadhab*, *hadh*, *hadhb*, *acaa1*, *acaa2*, *CABZ01065076.1*, *acat2*, *eci1*, *eci2*, *aldh3a2b*, *CAB Z01032488.1*, and *aldh9a1b*. Green indicates down-regulated genes.

**Figure 6 biology-10-00092-f006:**
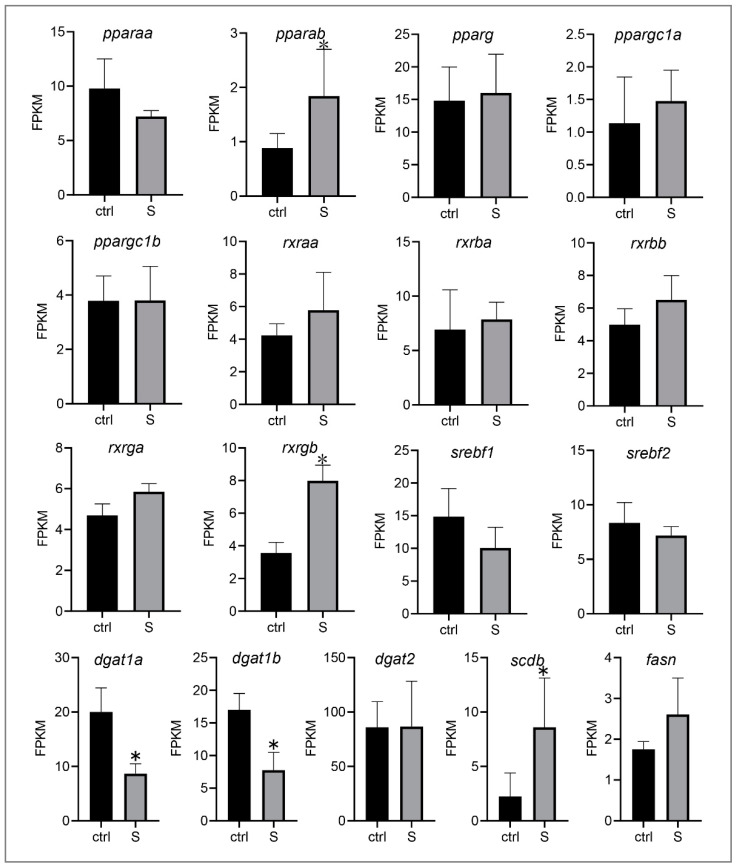
Effect of starvation on the gene expression involved in lipogenesis in liver tissues. Differential expression of *pparaa*, *pparab*, *pparg*, *ppargc1a*, *ppargc1b*, *srebf1*, *srebf2*, *dgat1a*, *dagt1b*, *dgat2*, *scdb*, *fasn*, *rxraa*, *rxrba*, *rxrbb*, *rxrga*, and *rxrgb* in the control and starved larvae. The FPKM value represents the mean ± SEM of three independent RNA-seq biological replicates. * indicates significant differences (*p* < 0.05).

**Figure 7 biology-10-00092-f007:**
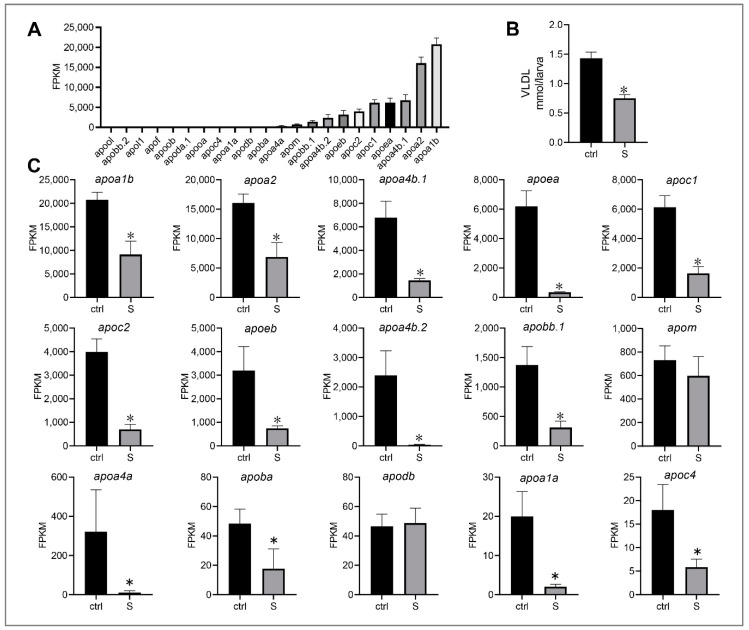
Effect of starvation on the gene expression involved in lipid transport in liver tissues. (**A**) Expression of apolipoprotein genes, including *apoa1b*, *apoa2*, *apoa4b.1*, *apoea*, *apoc1*, *apoc2*, *apoeb*, *apoa4b.2*, *apobb.1*, *apom*, *apoa4a*, *apoba*, *apodb*, *apoa1a*, *apoc4*, *apooa*, *apoda.1*, *apoob*, *apof*, *apol1*, *apobb.2*, and *apool*, were extracted in the control larvae from RNA-seq data. (**B**) Hepatic very low-density lipoprotein (VLDL) levels in the control and starved larvae at 8 dpf were determined. (**C**) Differential expression of the top 10 apolipoprotein genes in the control and starved larvae. The FPKM value represents the mean ± SEM of three independent RNA-seq biological replicates. * indicates significant differences (*p* < 0.05).

**Figure 8 biology-10-00092-f008:**
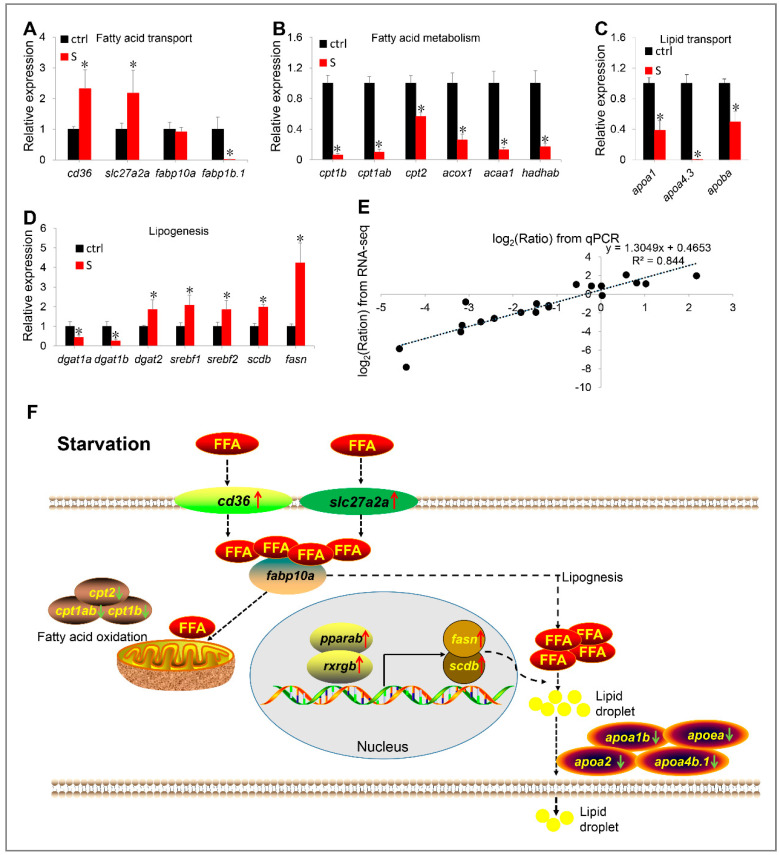
(**A–E**) Validation of RNA-seq data by qPCR. The mRNA levels of 20 genes involved in fatty acid transport (**A**), fatty acid metabolism (**B**), lipid transport (**C**), and lipogenesis (**D**) were measured by qPCR. (**E**) Correlation analysis of gene expression between RNA and qPCR. Scatter plot of gene expression in RNA-seq (y-axis) and qPCR (x-axis). “Ratio” in log_2_ (Ratio) represents the fold change with gene expression. * indicates significant differences (*p* < 0.05). (**F**) Schematic diagram of starvation triggering hepatic steatosis by regulating liver fatty acid intake, β oxidation, de novo lipogenesis and lipid transport. Starvation up-regulated transcription of *slc27a2a*, *slc27a6-like*, and *cd36*, promoting extrahepatic fatty acid uptake. In liver, starvation induces the expression of lipogenesis-related genes (*srebf1*, *srebf2*, *fasn*, and *scdb*) by activating the PPARα/RXR pathway, thereby promoting lipogenesis. Meanwhile, the fatty acid metabolism pathway was inhibited. Thus, most ingested extrahepatic fatty acids are used for lipogenesis rather than β oxidation. However, synthetic fat appears to be blocked in the liver, because starvation inhibits the expression of numerous apolipoprotein genes. Red arrows indicate up-regulated genes, green arrows indicate down-regulated genes. FFA: free fatty acid.

**Figure 9 biology-10-00092-f009:**
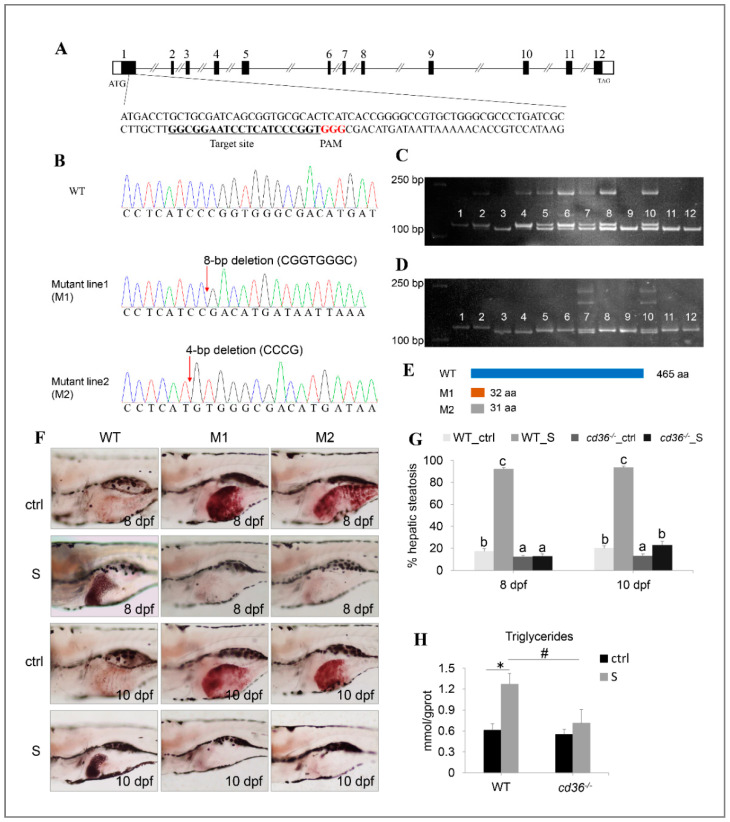
Establishment of the *cd36* knockout zebrafish line. (**A**) Genomic structure of the zebrafish *cd36* gene and CRISPR/Cas9 target site design. (**B**) DNA sequencing confirmed that two mutants (c. 86_93delCGGTGGGC, p.Pro29Argfs5) and (c. 84_87delCCCG, p. Ile28Metfs5) were generated. (**C**,**D**) Genotypes of *cd36* mutants were screened by the PAGE technology. Image C labels the mutant line1 (M1): Lanes 3, 9, 11, and 12 were identified as *cd36* homozygous mutant line; lanes 5, 6, 7, 8, and 10 were identified as *cd36* heterozygous mutant line; lanes 1, 2, and 4 were identified as WT zebrafish. Image D labels the mutant line2 (M2): Lanes 3 and 4 were identified as *cd36* homozygous mutant line; lanes 7 and 10 were identified as *cd36* heterozygous mutant line; lanes 1, 2, 5, 6, 8, 9, 11, and 12 were identified as WT zebrafish. (**E**) A diagram representative of WT and two truncated mutant CD36 proteins (M1 and M2). (**F**) Oil Red staining of WT and *cd36^−/−^* mutant zebrafish larvae at 8 and 10 dpf. The ORO-stained signal in intestinal lumen is the lipid droplets of exogenous nutrients. (**G**) Percentages of WT and *cd36^−/−^* mutants (M1) with hepatic steatosis at 8 and 10 dpf. Significant differences are marked by different letters. (**H**) TAG levels in starved WT and *cd36^−/−^* mutants (M1) at 8 dpf. # *p* < 0.05 compared to WT_S; * *p* < 0.05 compared to WT_ctrl.

**Table 1 biology-10-00092-t001:** Statistical analysis of sequencing data.

Samples	Ctrl1	Ctrl2	Ctrl3	Starve1	Starve2	Starve3
Raw reads (M)	50.82	45.52	54.45	44.12	46.26	44.79
Clean reads (M)	50.44	45.20	52.01	43.61	45.83	44.29
Good reads (%)	99.25	99.30	95.51	98.84	99.07	98.88
Error rate (%)	0.0250	0.0250	0.0251	0.0253	0.0251	0.0250
Q20 (%)	98.02	97.99	97.96	97.89	97.96	98.02
Q30 (%)	94.21	94.12	94.08	93.92	94.05	94.18
Total mapped (M)	46.18	41.18	47.85	39.87	41.96	40.68
Total mapped (%)	91.55	91.11	92.0	91.41	91.55	91.86
Uniquely mapped	39.27	34.77	40.42	33.15	35.03	34.30
Uniquely mapped (%)	77.85	76.92	77.71	76.02	76.42	77.44

## Data Availability

The data presented in this study are available on request from the corresponding author.

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
