# Peer review of "A Model Construction of Starvation Induces Hepatic Steatosis and Transcriptome Analysis in Zebrafish Larvae"

_biology, 2021, doi:10.3390/biology10020092_

Round 1

Reviewer 1 Report

The authors Hao Xu et al claim that starvation induces NASH in model of Zebrafish larvae, caused by dysregulated lipid metabolism. Also found an up-regulation of CD36 expression in the liver of zebrafish larvae. In order justify the CD36 role in NASH, authors generated the CD36 mutant line using CRISPR/Cas9 system and identified as a candidate gene that is playing a potential role in abnormal lipid metabolism in liver of Zebrafish larvae, a starvation induced NASH model. However, authors need to address the fallowing major and minor comments in order to be accepted this manuscript in this Biology journal, corresponding to a peer reviewed journal of MDPI.

Major comments:

  1. Figure 1D, Oil-Red-O Image quality is very poor and very hard to see the staining with low magnification. In addition to low magnification Images, provide the high magnification image from one the day of postfertilization (dpf) that can allow the readers to see the lipid droplets in starved zebrafish larvae.
  1. Figure 1D, I believed that the Oil Red-O staining was performed in whole body of Zebrafish larvae. The Image was not clearly indicating that the positive staining is in liver? Authors should include the Oil-Red-O stained images on frozen liver sections, which should indicate the deposition of Lipid droplets, specifically in liver of zebrafish larvae.
  1. In Figure 1E, authors showed increased liver Triglycerides (TG) in starved larvae at 8-days postfertilized (8-dpf) larvae. However, % of steatosis and Oil-Red-O Images showed at different days of postfertilization. Do you see any time dependent increased in liver TG under starved condition? If yes, please indicate the Data of liver TG at all the days (6-10 dpf) of postfertilized larvae and if not, need to elucidate.
  1. Do you see any microvesicular steatosis or lipid vacuoles? If yes, please provide the H&E stained images. If not please explain.
  1. In Figure 2, the transcriptome data analysis was indicating DEGs by using KEGG analysis. KEGG enrichment analysis indicating the several pathways involved and regulated. However, data did not specify the list genes involved in those each specific pathway. Therefore, authors must provide as a supplementary data indicating the list of regulated genes in those each specific pathways belongs too.
  1. Based on one of previous publication (Qilin Gu et al Hepatology 2014), some of the amino acid transporters known to be involved in the development of NASH under fasting condition. Authors did not show any data related to regulation of aminoacid transporters under starved stress condition. Have authors looked at regulation of any aminoacid transporters at the gene or protein levels in your stress induced NASH model of larvae? and what is the impact of Slc7a1, Slc7a2 and Slc7a3a in your model.
  1. Authors found that, dysregulated lipid metabolism under starved stress condition of NASH model and identified the CD36 as potential candidate. However, Slc27a2a has shown to be highest or similar levels of expression as CD36 in liver. Is there any presence of compensatory mechanism existed in the absence of CD36 in zebrafish larvae? However, authors did not show any data related to the expression of Slc27a2a and Slc27a6-like transporters in the liver of CD36 Knockout Zebrafish line?  
  1. In Figure 9H, I am surprised that the CD36 KO line showed similar TG levels or almost comparable TG levels between WT and KO line under starved condition? In addition, what is status of other major transporter Slc27a2a?
  1. Do authors look into any genes involved in denovo fatty acid synthesis and beta oxidation were increased or decreased in the absence of CD36?
  1. Do you propose any sensing mechanism that lead to increased expression of both CD36 and Slc27a2a under starved stress condition? Or any previous papers commented? Please comment and refer in the discussion which could elaborate potential insight for the readers.

MINOR COMMENTS:

  1. In Material and Methods section 2.1, it would be easier for readers if authors can indicate that fasting condition of Larvae and terminal time point of the experiment. Please rewrite the sentence.
  2. In Figure 1, Authors do not need to split the Figure 1F & 1G as the 2-pannels belong to one experiment. I suggest to indicate as one figure as control vs starved (S).
  3. In Figure 2C, authors indicated that the 28-genes expressed only in control and 9-genes were in starved larvae alone. Please list the gene names.
  4. In Figure 3, 4, 6 & 7, what is FKPM on the y-axis? I did not see anywhere in the manuscript mentioned or abbreviating of FKPM?
  5. Apparently it is very confusing about the gene expression data figures. One of the figures indicated data is from RNA-sey and other data was not notified clearly. For example, In Figure 3B and Figure 4B, Is the gene expression data performed from RNA seq or qRT-PCR? Please indicate clearly in the figure legend.
  6. Similar to my previous comment, In Figure 6 and Figure 7B, Is the gene expression data performed from RNA seq or qRT-PCR? Please indicate clearly in the figure legend.

Author Response

Dear Reviewer:

Thanks you for your comments and constructive suggestions for revising and improving our manuscript “A Model Construction of Starvation Induces Hepatic Steatosis and Transcriptome Analysis in Zebrafish Larvae”. We have carefully considered your comments, and have modified the manuscript accordingly. Your comments encourage us to further explore the pathogenesis of starvation-induced NAFLD in zebrafish. If there are other aspects in the manuscript that require further clarification, kindly let us know and we would be delighted to comply with.

Thanks again for all the helpful comments you provided.

Sincerely yours,

Hao Xu, Yu Jiang, Xiao-Min Miao, Yi-Xi Tao, Lang Xie and Yun Li*

Response to Reviewer Comments

Major concerns:

Point 1: Figure 1D, Oil-Red-O Image quality is very poor and very hard to see the staining with low magnification. In addition to low magnification Images, provide the high magnification image from one the day of postfertilization (dpf) that can allow the readers to see the lipid droplets in starved zebrafish larvae.

Response 1: Thanks for your suggestion. We have corrected and added the high magnification images in Figure 1D.

Point 2: Figure 1D, I believed that the Oil Red-O staining was performed in whole body of Zebrafish larvae. The Image was not clearly indicating that the positive staining is in liver? Authors should include the Oil-Red-O stained images on frozen liver sections, which should indicate the deposition of Lipid droplets, specifically in liver of zebrafish larvae.

Response 2: Thanks for your suggestion. We can be sure that the positive staining is the liver of zebrafish. On the one hand, we refer to the extensive literature on zebrafish liver Oil-Red-O staining. (e.g., HEPATOLOGY 2014;60:1581-1592; HEPATOLOGY 2014;60:1929-1941). On the other hand, we can provide the whole-mount in situ hybridization (WISH) of fabp10a, which is specifically expressed in the liver of zebrafish larvae and is a marker gene for liver (Figure a). Obviously, the site of Oil-Red-O staining was completely consistent with the site of fabp10a expression. Therefore, it can be determined that the positive staining is in the liver of zebrafish.

Regarding frozen liver sections, we do agree with reviewer’s views. However, zebrafish larvae are very small individuals (about 4 mm in body length at 5~8 dpf) and the liver tissue is very thin, thereby freezing sectioning is very difficult. We tried to freeze liver sections several times but failed. Of course, we are very willing to make further efforts to try to freezing liver sections, if
the reviewer is still not satisfied with our revised results.

Figure a

Point 3: In Figure 1E, authors showed increased liver Triglycerides (TG) in starved larvae at 8-days postfertilized (8-dpf) larvae. However, % of steatosis and Oil-Red-O Images showed at different days of postfertilization. Do you see any time dependent increased in liver TG under starved condition? If yes, please indicate the Data of liver TG at all the days (6-10 dpf) of postfertilized larvae and if not, need to elucidate.

Response 3: Thanks for your professional question. Yes, we observed an increase in liver TG over time under starvation conditions. The levels of TG reaches its maximum at 8 dpf. We have corrected and added detailed data in Figure 1E.

Point 4: Do you see any microvesicular steatosis or lipid vacuoles? If yes, please provide the H&E stained images. If not please explain.

Response 4: Thanks for your professional question. We only observed a large accumulation of lipid droplets in the liver of the starved zebrafish larvae (Figure b). With respect to microvesicular steatosis or lipid vacuoles, we have not observed. We speculate that short-term and simple starvation may not trigger this symptom. Of course, we will combine starvation with other treatments to further study in the following research.

Figure b

Point 5: In Figure 2, the transcriptome data analysis was indicating DEGs by using KEGG analysis. KEGG enrichment analysis indicating the several pathways involved and regulated. However, data did not specify the list genes involved in those each specific pathway. Therefore, authors must provide as a supplementary data indicating the list of regulated genes in those each specific pathways belongs too.

Response 5: Thanks for your suggestion. We have provided detailed supplementary data in Table S3.

Point 6: Based on one of previous publication (Qilin Gu et al Hepatology 2014), some of the amino acid transporters known to be involved in the development of NASH under fasting condition. Authors did not show any data related to regulation of amino acid transporters under starved stress condition. Have authors looked at regulation of any amino acid transporters at the gene or protein levels in your stress induced NASH model of larvae? and what is the impact of Slc7a1, Slc7a2 and Slc7a3a in your model.

Response 6: Thanks for your professional question. This article “Genetic Ablation of Solute Carrier Family 7a3a Leads to Hepatic Steatosis in Zebrafish During Fasting” is very interesting, and it is also meaningful for understanding the pathogenesis of starvation-induced NASH. In our study, however, we don’t seem to observe a significant change in the expression of the amino acid transporters, including slc7a1, slc7a2, slc7a3a, slc7a3b, slc7a4, slc7a5, slc7a6, under starvation stress (Figure c).

Figure c

Point 7: Authors found that, dysregulated lipid metabolism under starved stress condition of NASH model and identified the CD36 as potential candidate. However, Slc27a2a has shown to be highest or similar levels of expression as CD36 in liver. Is there any presence of compensatory mechanism existed in the absence of CD36 in zebrafish larvae? However, authors did not show any data related to the expression of Slc27a2a and Slc27a6-like transporters in the liver of CD36 Knockout Zebrafish line? 

Response 7: Thanks for your professional question. We detected the effect of CD36 knockout on the expression of slc27a2a and slc26a6-like in zebrafish liver. However, no significant changes in their expression were observed from our qPCR data. Thus, we speculate that there is no presence of compensatory mechanism existed in the absence of CD36 in zebrafish larvae. In our study, the role of CD36 is irreplaceable by slc27a2a in the process of starvation-induced hepatic steatosis in zebrafish.

Point 8: In Figure 9H, I am surprised that the CD36 KO line showed similar TG levels or almost comparable TG levels between WT and KO line under starved condition? In addition, what is status of other major transporter Slc27a2a?

Response 8: Yes, we detected a downward trend in liver TG levels of CD36 KO line compared to wild-type zebrafish under normal conditions, but it is not significant (p > 0.05). In our study, the expression of slc27a2a in CD36 KO line was not affected. We believe that CD36 KO zebrafish can mediate the transmembrane transport of slc27a2a to promote the uptake of extra-hepatic fatty acids. However, slc27a2a may not compensate for the role of CD36 in uptaking extrahepatic fatty acids under starvation conditions. In our next work, we will establish the slc27a2a KO line and reveal the role of this gene in starvation-induced zebrafish hepatic steatosis.

Point 9: Do authors look into any genes involved in denovo fatty acid synthesis and beta oxidation were increased or decreased in the absence of CD36?

Response 9: Thanks for your professional question. In this work, we only preliminarily revealed the importance of cd36 in regulating starvation-induced hepatic steatosis in zebrafish larvae. Regarding the effect of cd36 knockout on zebrafish hepatic lipid metabolism (e.g., denovo fatty acid synthesis and beta oxidation), this is a question we will focus on in the following research.

Point 10: Do you propose any sensing mechanism that lead to increased expression of both CD36 and Slc27a2a under starved stress condition? Or any previous papers commented? Please comment and refer in the discussion which could elaborate potential insight for the readers.

Response 10: Compared with CD36 gene, the study of slc27a2a is relatively less. At present, the study of slc27a2a in fish has been limited to gene expression. Following the suggestions of reviewers. We have added new comments and references in the Discussion section of the article.

MINOR COMMENTS:

Point 1: In Material and Methods section 2.1, it would be easier for readers if authors can indicate that fasting condition of Larvae and terminal time point of the experiment. Please rewrite the sentence.

Response 1: Following the suggestions of reviewers, we have corrected our description in Material and Methods section 2.1.

Point 2: In Figure 1, Authors do not need to split the Figure 1F & 1G as the 2-pannels belong to one experiment. I suggest to indicate as one figure as control vs starved (S).

Response 2: Thanks for your suggestion. We have corrected it.

Point 3: In Figure 2C, authors indicated that the 28-genes expressed only in control and 9-genes were in starved larvae alone. Please list the gene names.

Response 3: Thanks for your suggestion. We have added it in Table S2.

Point 4: In Figure 3, 4, 6 & 7, what is FKPM on the y-axis? I did not see anywhere in the manuscript mentioned or abbreviating of FKPM?

Response 4: Thanks for your question. In the RNA-seq data, the gene expression levels were calculated using the method of fragments per kilobase of exon per million mapped reads (FPKM). Following the suggestions of reviewers, we have corrected and added its definition in each legend and Material and Methods section 2.6.

Point 5: Apparently it is very confusing about the gene expression data figures. One of the figures indicated data is from RNA-sey and other data was not notified clearly. For example, In Figure 3B and Figure 4B, Is the gene expression data performed from RNA seq or qRT-PCR? Please indicate clearly in the figure legend.

Response 5: Thanks for your suggestion. The gene expression data in Figure 3-7 is derived from the RNA-seq data. We have corrected and added data information in the figure legend.

Point 6: Similar to my previous comment, In Figure 6 and Figure 7B, Is the gene expression data performed from RNA seq or qRT-PCR? Please indicate clearly in the figure legend.

Response 6:  Following the suggestions of reviewers, we have added data information in the figure legend.

Reviewer 2 Report

The manuscript by Xu et al provides mechanistic insight into an intriguing, but understudied disease: starvation-induced NAFLD. The authors established a model of starvation-induced hepatic steatosis, performed RNAseq and validated the results, identified CD36 as a critical player in this process, as well as made interesting observations about the pathways involved in this process. Minor changes and additions would improve the clarity of the manuscript and interest to readers. 

  1. The authors should comment on the relative prevalence of primary NAFLD vs secondary NAFLD in order to put the significance of secondary liver disease in perspective.
  2. How do the differentially regulated genes in the starvation-induced hepatic steatosis model compare to zebrafish models where a high fat or high cholesterol diet is used to recapitulate primary NAFLD?
    1. Comparing the up-regulated genes in the model presented here and publically available data sets of primary NAFLD in zebrafish could provide insights that would interest a broad readership.
    2. The authors state at the end of the discussion that the ‘CD36 pathway plays a crucial role in NAFLD induced by high-fat diet’ but don’t provide further information. Additional discussion, either as rationale for looking at CD36 in the results, or in the discussion would improve the clarity of the manuscript.
  3. Figure 5 is difficult to comprehend. It would make the diagram clearer to use the names of the genes where they act in the fatty acid metabolism pathway, rather than numbers assigned by the program. Potentially, simplification of the figure, indicating the steps/processes where the DEGs act, but greatly improve the clarity of this figure.

Author Response

Dear Reviewer,

We really appreciate all the valuable comments you have provided, which are very helpful for revising and improving our manuscript “A Model Construction of Starvation Induces Hepatic Steatosis and Transcriptome Analysis in Zebrafish Larvae”. We have considered the comments carefully and have made revision which marked in red in the revised manuscript. We have tried our best to revise our manuscript according to the comments. The points raised by you have been dealt with in the revised manuscript, as detailed in the Author's Notes to Reviewer. We would like to take this opportunity to express our gratitude to you in helping us to clarify a few points in our revised manuscript. Your constructive comments have enabled us to arrive at an improved manuscript. If there are other aspects in the manuscript that require further clarification, kindly let us know and we would be delighted to comply with.

Thanks again for all the helpful comments you provided.

Sincerely yours,

Hao Xu, Yu Jiang, Xiao-Min Miao, Yi-Xi Tao, Lang Xie and Yun Li*

Response to Comments

Point 1: The authors should comment on the relative prevalence of primary NAFLD vs secondary NAFLD in order to put the significance of secondary liver disease in perspective.

Response 1: Thanks for your suggestion.  From the literature we have reviewed, there are very few studies on secondary NAFLD. Currently, there are no detailed data on the relative prevalence of primary and secondary NAFLD, but NAFLD caused by malnutrition and dieting is not uncommon. Therefore, we suggest that this study is of great significance to the development of secondary NAFLD research. Following the suggestions of reviewers, we added new comments in Introduction.

Point 2: How do the differentially regulated genes in the starvation-induced hepatic steatosis model compare to zebrafish models where a high fat or high cholesterol diet is used to recapitulate primary NAFLD?

  1. Comparing the up-regulated genes in the model presented here and publically available data sets of primary NAFLD in zebrafish could provide insights that would interest a broad

Response 2.1: Thanks for your suggestion. We compared our data with zebrafish models on high fat or high cholesterol diet (Chen Bo et al., frontiers in Endocrinology 2018). We observed that the two treatments had significantly different effects on gene expression in zebrafish liver. In the model of high fat or high cholesterol diet, glucose metabolism (i.e., irs2, glut1, glut2 and pepck), lipid oxidation (i.e., ppara and cpt1a), and ER stress genes (i.e., ddit3 and grp78), antioxidant gene (i.e. gpx1a) were up-regulated significantly. However, in our study, lipid oxidation (e.g., cpt1b, cpt1ab and cpt2), glucose metabolism (e.g., gk3p, adpgk and hkdc1), and antioxidant genes (e.g., gpx1b, gpx4a, cat, prdx1 and txn) were down-regulated. Clearly, starvation stress inhibits energy expenditure as well as the tissue's antioxidant defenses. In addition, in the model of high fat or high cholesterol diet, autophagy in liver was affected significantly. In our study, however, no significant change was observed in the autophagy pathway.

      Regarding lipogenesis, we found that the two treatments have certain similarities in gene expression, despite the differences in genes. In the model of high fat or high cholesterol diet, lipogenic genes (e.g., cidec and chrebp) were up-regulated. In the starvation model that we studied, lipogenic genes (e.g., dgat2, srebf1, srebf2, scdb and fasn) were up-regulated. Totally, it is of great significance to compare and analyze the NAFLD induced by the two treatments. In the following research, we will design rigorous experiments for further study.

  1. The authors state at the end of the discussion that the ‘CD36 pathway plays a crucial role in NAFLD induced by high-fat diet’ but don’t provide further information. Additional discussion, either as rationale for looking at CD36 in the results, or in the discussion would improve the clarity of the manuscript.

Response 2: Following the suggestions of reviewers. We have added the CD36 information at the end of the Discussion. In our manuscript, we point out that CD36 is a key transporter of free fatty acid uptake in liver. In high-fat diet induced NAFLD patients and animal models, up-regulation of CD36 were commonly detected. On the contrary, suppressing its expression can effectively improve hepatic lipid accumulation.

Point 3: Figure 5 is difficult to comprehend. It would make the diagram clearer to use the names of the genes where they act in the fatty acid metabolism pathway, rather than numbers assigned by the program. Potentially, simplification of the figure, indicating the steps/processes where the DEGs act, but greatly improve the clarity of this figure.

Response 3: Thanks for your suggestion. We have provided a high-definition image. After zooming in, the new Figure 5 can be clearly presented, and the names of the DEGs have been added to the corresponding locations.

Round 2

Reviewer 1 Report

Dear Authors,

Thank you for the comprehensive responses and additional information along with Images that was provided. The current Manuscript version looks good for me with all the information needed to support the hypothesis. This Manuscript acceptable with its current form. congratulation to authors.